# Modifying and validating the social responsiveness scale edition 2 for use with deaf children and young people

**Barry Wright** [1]*, **Helen Phillips**[2], **Ann Le Couteur**[3], **Jennifer Sweetman**[2], **Rachel Hodkinson** [2], **Amelia Ralph-Lewis**[4], **Emily Hayward** [2], **Alice Brennan**[2], **Josie Mulloy**[5], **Natalie Day**[2], **Martin Bland**[1], **Victoria Allgar**[1]

**1** University of York, York, England, **2** Leeds and York Partnership NHS Foundation Trust, York, England, **3** Newcastle University, Newcastle, United Kingdom, **4** South London and Maudsley NHS Foundation Trust, London, United Kingdom, **5** Surrey and Borders Partnership NHS Foundation Trust, Leatherhead, United Kingdom

* barry.wright1@nhs.net

## Abstract

A Delphi consensus methodology was used to adapt a screening tool, the Social Responsiveness Scale– 2 (SRS-2), for use with deaf children including those whose preferred communication method is sign language. Using this approach; 27 international experts (The Delphi International Expert Panel), on the topic of autism spectrum disorder (ASD) in deaf people, contributed to the review of item content. A criterion for agreement was set at 80% of experts on each item (with 75% acceptable in the final fourth round). The agreed modifications are discussed. The modified SRS-2 research adaptation for deaf people (referred to here as the "SRS-2 Deaf adaptation") was then translated into British Sign Language using a robust translation methodology and validated in England in a sample of 198 deaf children, 76 with Autism Spectrum Disorders (ASD) and 122 without ASD. The SRS-2 Deaf adaptation was compared blind to a NICE (National Institute for Health and Care Excellence) guideline standard clinical assessment. The area under the Receiver Operating (ROC) curve was 0.811 (95% CI: 0.753, 0.869), with an optimal cut-off value of 73, which gave a sensitivity of 82% and a specificity of 67%. The Cronbach Alpha coefficient was 0.968 suggesting high internal consistency. The Intraclass Correlation Coefficient was 0.897, supporting test-retest reliability. This performance is equivalent to similar instruments used for screening ASD in the hearing population.

## Introduction

To date, there are no published Autism Spectrum Disorder (ASD) screening or diagnostic assessment tools that have been validated for use with deaf people [1]. Most current measures assume individuals can easily access spoken language [1,2] and often include questions or tasks that are inappropriate for use with deaf people, such as whether the person responds to their name being called [3]. Screening instruments and diagnostic interviews also ask questions

**Funding:** This research is funded by Medical Research Council as part of the Diagnostic Instruments for Screening Deaf Children Study (DIADS) - BW, AL-C, MR/K015435/1. The funders had no role in study design, data collection and analysis, decision to publish, or preparation of the manuscript. https://gtr.ukri.org/projects?ref=MR%2FK015435%2F1.

**Competing interests:** The authors have declared that no competing interests exist.

such as whether the parent/carer has ever considered that the child/young person might be deaf [4]. Therefore items such as these cannot be used to discriminate between ASD and non-ASD in a deaf person. There is an urgent need to modify existing ASD assessment procedures and/or develop more appropriate measures for use in screening and diagnosis in the deaf population.

The term 'deaf' in this paper includes individuals who are mild, moderate, severe or profoundly deaf [5] and may use a range of audiological aids or communication modalities (such as a signed language or spoken language).

There are estimated to be over 45,000 deaf children living in the United Kingdom (UK) [6] with 40% of these expected to have additional needs [7]. No prevalence study has been undertaken in the UK to identify the number of deaf children with ASD. One large parent report cohort carried out in the United States (US) in 2009/10 estimated that 1.7% of deaf children had received a diagnosis of ASD [8]. This work was conducted using parent report and may be less accurate than population research using direct assessments, albeit with the additional caveat that to date there are no validated assessment tools for use with deaf people. Using the 1.7% rate suggested by the Gallaudet Research Institute, the estimated number of children with possible ASD in the UK would be approximately 765.

The average age with which an ASD diagnosis is made is reported to be 55 months in the UK [9]. The equivalent statistic for a US study is reported as 3.1 years [10]. with another US study of about the same time suggesting that deaf children receive a diagnosis at a mean age of 15 years (range 5–16 years) [11]. Although this was a small sample, it highlights the point that many deaf children are receiving a diagnosis too late to plan helpful education and parental support. Late diagnosis may be related to inexperienced assessment of deaf children in a hearing environment; misunderstanding of behaviours associated with language deprivation, limited availability of expertise for assessment or reduced numbers of families coming forward for assessment or support. Fiorillo [12] for example, reported that when US children demonstrated unusual behaviours 8% of parents of hearing children requested referral to child mental health services compared with none of the parents of deaf children.

Over 90% of deaf children are born to hearing parents [13], most of whom are not expecting a deaf infant. They are likely to be on a steep learning curve in understanding the deaf lived experience. Hearing parents are often relatively unprepared for the necessary language and education choices, the myriad of possible educational and social challenges and the range of possible life trajectories available for their deaf children. Deaf children with limited access to language in early life are more likely to experience delayed 'theory of mind' development [14], a recognised feature of ASD. This feature can lead to delays in both the development of empathy and socio-emotional skills in deaf children, which can masquerade as ASD [15]. All these factors add challenges to the ASD diagnostic assessment process.

Some aspects of ASD can present differently in hearing children compared to deaf children, for example, pronominal reversal is seen in the former group, but not the latter [16]. These differences in ASD presentations in deaf children are not well known clinically [17]. Children may be considered to have traits of ASD if they seem unable to interact in groups; however, deaf children may experience significant interactional challenges in noisy group settings and this may be misunderstood by those observing them [18]. These clinical complexities mean that clinicians in both child health and child mental health services may find the ASD diagnostic assessments and differential diagnoses of deaf children challenging [17]. Furthermore, there are only a limited number of professionals with expertise in both ASD and working with deaf people [19]. Indeed parents of deaf children regularly report experiencing considerable barriers on the route to a completed assessment [20]. These barriers include delays in accessing referral for an ASD assessment and a range of challenges in the assessment process in relation

to communication such as not having a language match between clinician and child and the challenges created when working with an interpreter to administer assessment [21].

For these reasons, appropriate and validated screening tools, as well as ASD diagnostic assessment tools need to be available for deaf children. A UK National Deaf Child and Adolescent Mental Health Service (NDCAMHS) was launched in 2009 [22] following a successful pilot service with independent evaluation undertaken by the Social Policy Research Unit (SPRU), University of York [23]. One of the clinical services provided by NDCAMHS is specialist support for complex presentations of ASD and second opinion assessments for deaf children and young people [24]. Reviewing the referrals, to this second opinion service, further highlights that to improve access and accuracy of ASD diagnosis in deaf children there is an urgent need for high quality, validated screening and diagnostic assessment tools. This will make accurate and reliable assessments available in community child health and child and adolescent mental health settings. To address this need we set out to adapt and validate for use with deaf children an ASD screening tool, the Social Responsiveness Scale (SRS-2) [25] and two ASD diagnostic assessment tools, which were a parent semi-structured interview the Autism Diagnostic Interview-Revised (ADI-R) [26] and a play/interaction based assessment the Autism Diagnostic Observation Schedule second edition (ADOS-2) [27]. We report here the research involving the adaptation, translation and validation of the SRS-2.

The SRS-2 is a screening questionnaire with 65 items that focuses on behaviours and symptoms seen in ASD. It is completed by an adult (parent, relative or teacher) who knows the child well. Responses are based on the child's behaviour for the previous 6 months. It has been validated for use in the general population in several different countries including the UK [25,28,29]. The SRS-2 generates total scores, which are derived from the scores for five Subscales: Social Awareness, Social Cognition, Social Communication, Social Motivation and Restricted Interests and Repetitive Behaviour [25,30]. Studies show that the SRS-2 can be used to discriminate both within the autism spectrum and between ASD and other disorders, which makes the questionnaire useful for differential diagnosis [31]. The total score is converted to a T-score to determine whether the individual meets the most current diagnostic criteria for ASD [28].

This study included an international online Delphi Consensus procedure that used expert opinions (from a range of stakeholders) and a pre-defined level of consensus agreement, to achieve a set of modifications for the written English SRS-2 Deaf adaptation version of the screening instrument.). This was followed by a translation (including a translation and independent blinded back translation) of the modified SRS-2 Deaf adaptation into British Sign Language (BSL; which has not been validated. A validation study of this newly modified instrument SRS-2 Deaf adaptation was then undertaken in a sample of deaf children and families recruited from across England.

## Materials and methods

### Approvals

The study was reviewed and approved by Research and Development at Leeds and York Partnership NHS Foundation Trust (LYPFT). A positive ethical opinion was obtained from National Research Ethics Service (NRES) Committee Yorkshire & the Humber—South Yorkshire. REC Reference: 15/YH/0093 for the Delphi Consensus Phase on 22/05/2014 and for the validation phase of the study on 17/04/2015. The study was undertaken with the agreement of the original authors of the SRS-2 [28] and the relevant permissions were obtained from Western Psychological Services (WPS), the publishers of the measure.

## The Delphi consensus method

Previous research has established that the Delphi consensus methodology is a reliable and valid structured communication technique for achieving consensus using an iterative procedure with a panel of experts. This process reduces the risk of undue individual influence whilst facilitating open sharing of opinions [32]. It has previously been successfully used to identify both ways in which assessment tools need to be revised [33] and the development of agreed sets of recommended modifications [34]. Using the Delphi consensus methodology, international experts in the field were invited through an internet platform to join a 'quasi-anonymous' Delphi International Expert Panel [35,36]. This procedure allowed relatively easy international access [34]. Experts were asked to give their opinions through a series of structured questions (see Fig 1) and the process was repeated in an iterative manner with a revised set of structured questions circulated to the experts over a total of four rounds [34,37]. For each item of the English version of the SRS-2, experts were invited to choose from three options. These were whether (i) the question should remain unchanged; (ii) the experts recommended modifications for use in a deaf population or (iii) the question should be rejected if in their opinion the item does not discriminate between the presence or absence of ASD in a deaf population. At the end of each round the researchers collated the responses and presented the findings to an Independent Research Review Team (details follow Fig 1). A reduced number of items including proposed amended items were then circulated to the international experts as a modified series of structured questions for the next round of the Delphi consensus. This process was repeated to achieve Delphi International Expert Panel group consensus according to the pre-specified criteria [32]. Items were banked as accepted or rejected if they met these criteria.

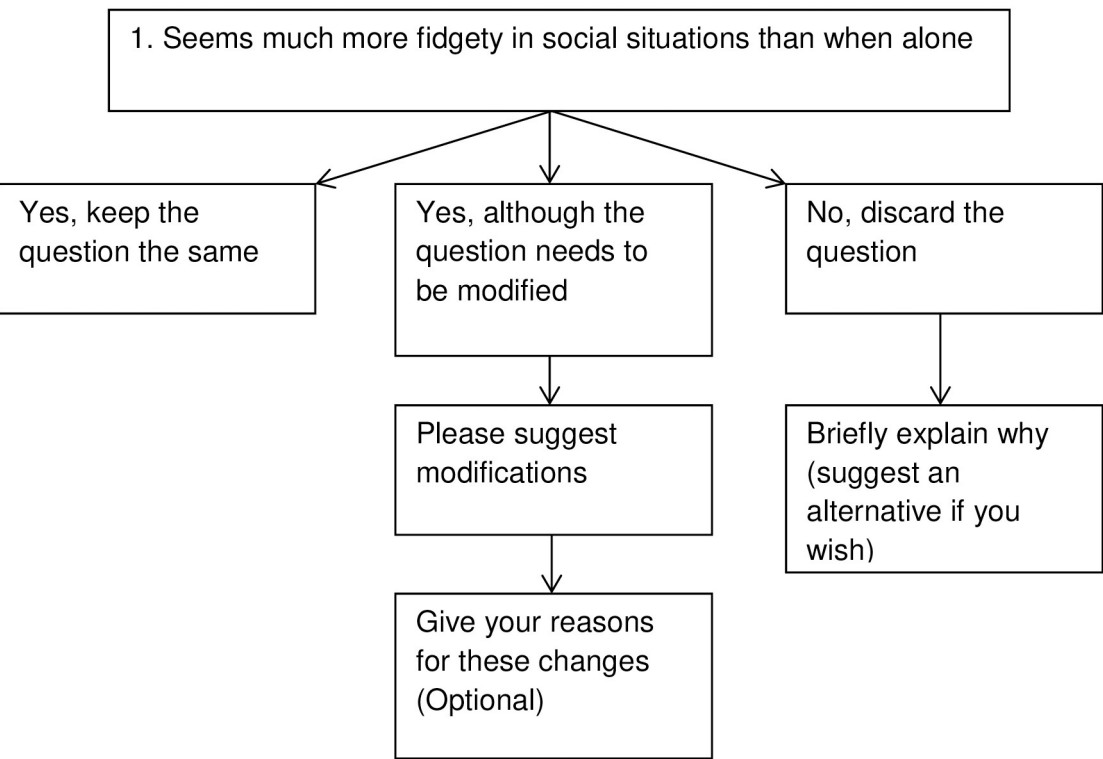

**Fig 1. Flowchart illustrating example response options presented with each SRS-2 item.**

**The delphi international expert panel members.**    Potential experts were recruited by contacting all known international teams with clinical expertise. They were identified through a scoping review of the literature and a search for clinical service websites. It was also announced at several international conferences. Individuals from those services, or the authors of relevant academic papers, were invited to participate by filling in a brief questionnaire about their expertise and experience. To be included, individuals were required to have: a minimum of one year's experience of working with or caring for deaf children with ASD, and to also have active experience using screening questionnaires for ASD with deaf children.

**Data collection methods used online survey facilities (SurveyMonkey).**    The recruited experts were given three response options for each question (see Fig 1). Where individual panel members indicated that questions needed to be modified they had the opportunity to make modification suggestions and reasons for needing to modify questions. Where possible, we also asked them to give any evidence for their opinions (e.g. a relevant publication reference). Some questions were deemed to be unsuitable to assess ASD in deaf children and were discarded. Where new/alternate questions were put forward by the expert panel members' evidence for their opinions was requested if possible.

## Independent research review team

An independent research review team was developed to discuss the collated International Expert Delphi Consensus findings from each iterative round. The team included: parents of deaf children with ASD, educationalists with experience of deaf children with ASD, together with deaf and hearing clinical and academic research practitioners working in child mental health fields and linguistics. It was also important for the team to include deaf professionals in order to ensure that the discussions incorporated accurate knowledge of deaf cultural perspectives.

## Procedure in the Delphi consensus for SRS-2

The collated findings at the end of each Delphi round were presented to the Independent Research Review Team. All SRS-2 items that achieved 80% agreement by the Delphi International Expert Panel were either accepted and 'banked' or rejected as appropriate. Each item with a level of agreement below 80%, together with the expert panel recommendations for modifications to the wording, was discussed by the Independent Research Review Team. The proposed revised wording was agreed before these items were recirculated to the expert panel for consideration in the next Delphi Consensus online round. The number of items included in each subsequent round was therefore reduced. This process was then repeated with items achieving over 80% agreement being accepted and further discussion occurring for those items that required further modification. For the fourth and final round, the pre-specified percentage agreement was 75%. The final written English version of the modified SRS-2 research adaptation for deaf people (which will be referred to here as the "SRS-2 Deaf adaptation") was then available, to be translated in British Sign Language (BSL). The BSL version was created using a high quality forward and backward blinded translation methodology [38].

One of the pre-specified goals for the researchers and the original authors was to try to ensure that the modifications of instrument content and any adjustments to the wording of items were kept to a minimum. This allowed as much conceptual integrity of the original instrument and scoring system to be retained as possible.

## Validation study methodology of the SRS-2 Deaf adaptation for use with deaf children and young people

### Participants and recruitment methods

Participants in this research included deaf children/young people with an ASD diagnosis and deaf children/young people without an ASD diagnosis and their parents/guardians. In order to meet inclusion criteria, children and young people needed to be between 2 and 18 years old and were required to have bilateral hearing loss of at least 40dBHL. Children were not excluded on the basis of learning disability or other health or mental health co-morbidities and were included whatever their attributions about ASD (e.g. ASD as part of human neurodiversity or as a condition). Methods of communications used by deaf children and young people, as well as their parents/guardians, included: spoken English, Sign Supported English (SSE) or British Sign Language (BSL).

Recruitment took place across England and involved contacting schools for the deaf, mainstream schools with specialist resource bases for deaf children and special educational needs schools. Schools were asked to circulate details of the study to potentially eligible children and their families, as well as attending information sessions at some schools. In addition to school-based recruitment, the study team contacted the ten NDCAMHS teams, as well as other CAMHS services across England. They were asked to forward study information to families who currently, or had previously, accessed these services and met the inclusion criteria. Throughout the research, the study team worked closely with the NDCAMHS teams as they carried out new ASD assessments and could therefore ask interested families to consent to be part of this study alongside their assessment process. Many organisations including the National Autistic Society, National Deaf Children's Society, and the national ASD-UK and Daslne (Database of Children Living with Autism Spectrum Disorder in the North East) research database agreed to share study information with their members. The study was also advertised on various social media platforms including a popular online blog aimed towards the Deaf community, and various online parenting groups and platforms. All families were given a participant information sheet and those that took part were contacted by a researcher who obtained written informed consent. For minors (those under 16 years of age) parents/guardians gave written informed consent and the child/young person gave assent before taking part.

The SRS-2 Deaf adaptation was completed by the parent/carer of the deaf child/young person with sequential parent/carers (a minimum of 80 based on statistical calculations) asked to fill in the questionnaire again after two weeks in order to calculate test-retest reliability.

### Diagnostic procedure

All recruited children/young people recruited to this study by their local clinical teams were also assessed independently by clinicians from the NDCAMHS with specialist expertise in the field of ASD in deaf children/young people. These clinicians had no previous knowledge of the child and/or family; they travelled nationally to complete these assessments, and were blind to the SRS-2 Deaf adaptation scores. These NICE guideline standard clinical assessments carried out by the senior multidisciplinary child mental health clinicians from NDCAMHS [22] were based on World Health Organization Research Diagnostic Criteria for ASD [39].

The NDCAMHS clinicians met with parents and children to gather a comprehensive history of the child's development, family and medical history. They also observed and interacted with children at home or school and were able to access information from a recently completed Social Communication Questionnaire (SCQ) [40]. Clinicians also had the option to

speak with a teacher to gather further information and view additional professional reports (such as speech and language therapy and educational psychology) where available. The NDCAMHS clinicians then amalgamated the information using a reporting matrix, which was based on the ICD-10 research diagnostic criteria for ASD [39], to determine whether a diagnosis of ASD was appropriate. The children were classified as having ASD or not having ASD based on the NICE guideline compliant clinical assessments (described above). For a small number of children this independent assessment was not available. In these cases where there was either a parent report, of an existing formal diagnosis for ASD by a professional NHS clinical assessment or a score above the upper threshold ($\geq 15$) on a recently completed SCQ, (although this measure has not been adapted or validated for use with deaf people, it is a widely used screening tool for ASD with established cut-offs) [4,40], they entered an ASD 'diagnostic group'. These were included in an additional sensitivity analysis comparing results with analysis using only those children who had a NICE guideline compliant independentclinical assessment.

## Sample size (validation)

The sample size of 65 per group was based on estimating the difference in mean scores between, deaf children with ASD and deaf children without ASD to within ±0.34 standard deviations (95% confidence interval on each side of the estimate). For the test-retest reliability of SRS-2-Deaf adaptation, a sample size of at least 80 children was chosen to achieve 80% power to detect an intraclass correlation of 0.70 with a significance level of 0.05.

## Analysis

Descriptive statistics are presented as mean (sd) or number (percentage). The STARD (STAndards of Reporting Diagnostic accuracy studies) 2015 flow diagram [41] is presented. The SRS-2 Deaf adaptation raw and T scores were compared between groups using a t-test. The SRS-2 Deaf adaptation T scores were compared against the diagnostic groups established, (as a sensitivity analysis based on the gold standard clinical interview only), using sensitivity and specificity which was calculated based on the published diagnostic cut-off values (T scores of 59 and below are considered to be within typical limits and generally not associated with clinically significant ASD). The success criteria and target values for the sensitivity and specificity, as specified in the research protocol, was 80% sensitivity and 70% specificity. Receiver Operator Curves (ROC) was used to calculate the Area Under the Curve (AUC) with 95% confidence intervals (CI) and to determine the optimal cut-off value, the value with highest Youden Index was used. The reliability of the scoring algorithms in deaf children, was explored using Cronbach's alpha [42]. The Intraclass correlation coefficient (ICC) was calculated to check the test-retest reliability for SRS-2-Deaf adaptation. The ICC estimates and their 95% confidence intervals were calculated based on absolute-agreement and a 2-way mixed-effects model [43]. Analysis was undertaken on STATA/SE 14.2 [44].

## Results

### Delphi consensus

150 invitation letters were sent to international experts with experience of ASD in deaf children and young people, from which 44 responded and were given a unique identifier code. 40 completed the demographic questionnaire. Of these, two were not eligible as they had no experience of working with deaf children with ASD, five decided not to participate, one did not have sufficient time to be involved, one did not feel they had sufficient clinical experience and

one individual did not complete the demographic questionnaire in time. Twenty seven international experts took part in the four rounds of the Delphi Consensus process for the SRS-2. As can be seen in Table 1, the majority of participants were from professional backgrounds (typically health or education), predominantly used spoken English as a preferred language, and were resident in developed nations.

Most experts were based in Australia, England and the United States. All were able to access the English materials and had experience of working with deaf children: 45% of respondents had between 1–10 years' experience, 20% had 11–20 years' experience and 30% of respondents had at least 21 years of experience working with deaf children. 80% had at least 1–5 years'

**Table 1. Demographic characteristics for the experts recruited to the Delphi Consensus process.**

| | Completed demographic questionnaire | Invited to review |
|---|---|---|
| | Delphi International Expert Panel Participants (%) | SRS-2 (%) |
| **Total** | 40 | 27 |
| **Background** | | |
| Parent/Carer of a deaf child with ASD (another parent was also a professional included in the row below) | 1 (2.5) | 1 (4) |
| Professional | 39 (97.5) | 26 (96) |
| **Gender** | | |
| Female | 34 (85) | 24 (89) |
| Male | 6 (15) | 3 (11) |
| **Hearing status** | | |
| Deaf | 4 (10) | 4 (15) |
| Hearing | 36 (90) | 23 (85) |
| **Preferred Language** | | |
| Spoken English | 38 | 24 (89) |
| Australian Sign Language (AUSLAN) | 3 | 3 (11) |
| American Sign Language (ASL) | 2 | 1 (4) |
| British Sign Language (BSL) | 2 | 2 (7) |
| Dutch | 1 | 1 (4) |
| **Country of current residence** | | |
| England | 15 | 8 (30) |
| Australia | 14 | 11 (41) |
| United States of America | 8 | 6 (22) |
| Netherlands | 2 | 1 (4) |
| Russia | 1 | - |
| **Occupation** | | |
| Psychologist | 11 (27.5) | 6 (22) |
| Psychiatrist | 8 (20) | 3 (11) |
| Speech & Language Therapist or Pathologist | 5 (12.5) | 4 (14) |
| Teacher of the Deaf | 4 (14) | 4 (14) |
| Educational Sign Language Model (teacher Aide or interpreter) | 3 (7.5) | 3 (11) |
| Developmental Paediatrician | 2 (5) | 1 (4) |
| Specialist Educational Advisor | 2 (5) | 1 (4) |
| Professor or research academic | 2 (5) | 2 (7) |
| Deaf Service consultant (child mental Health) | 1 (2.5) | 1 (4) |
| Program Manager | 1 (2.5) | 1 (4) |
| Trainee clinical psychologist | 1 (2.5) | 1 (4) |

experience working with children with ASD. 40% had not carried out any ASD assessments with children within the previous 5 years, 45% reported working with between 1–9 deaf children with ASD, 20% had worked with 10–19, and 30% with 20 or more. The remaining 5% did not have any recent experience working with deaf children with ASD, but had substantial past experience. All individuals had experience of using a screening tool or questionnaire when working with a deaf child with ASD. The roles of individuals who volunteered to review the SRS-2 included psychologists, teachers and speech and language therapists (see Table 1). One person indicated that they were both a parent/carer and a professional (in Table 1 they have been included as a professional).

**Amendments made through the Delphi consensus process.** Out of the original 65 items, two were discarded and the remaining 63 were kept. One new item was added; bringing the total number of items in the SRS-2 Deaf adaptation to 64.

The SRS-2 Deaf adaptation was agreed after four rounds of the Delphi process (see Fig 2).

As can be seen in Table 2, during the first round 16 items were agreed without any discussion. At the end of round 2, a further 11 original items were considered appropriate for use with deaf children without any changes and one new item was agreed as appropriate to include in the modified tool. Overall, a total of 27 items remained the same as the original items. Two items were discarded during round 1 and are described in the next section. The remaining 36 original items were amended between rounds 2 and 4 for several reasons which are discussed below. Some items were modified to reduce potential false positives; many deaf children have poor access to language and communication that can lead to language, social emotional and/or developmental delay. The SRS-2 versions for 4–18 years old and 30–54 months were merged, as it was deemed there was limited value in separating them, once the other changes had been made. This reflected the fact that the Delphi International Expert Panel group believed that the items in the 4–18 group after adaptation were appropriate for use in the younger age group. At the end of the four rounds, five items had not reached consensus. Following guidance from the instrument author, these five items were included in the modified instrument in their original form.

**Discarded items.** Two questions were discarded in Round 1:

Item 52

- *Knows when he or she is talking too loud or making too much noise,*

Item 53

- *Talks to people with an unusual tone of voice (for example, talks like a robot or like he or she is giving a lecture)*

Both differences in volume of speech and 'unusual' tone of voice can be seen in deaf children without ASD [45]. The experts indicated that since deaf people communicate in many different ways with or without use of vocal apparatus, it would be difficult to identify appropriate alternative concepts to replace these items.

**Modified items.** The main modifications agreed can be organised into three categories: the communication environment, the experience of being a deaf person and structural modifications.

## The communication environment

The Delphi experts agreed that the answers to some questions would be different depending on the communication environment (i.e. the context in which the child was being seen). For example, when a typically developing deaf child/young person is in a situation with people

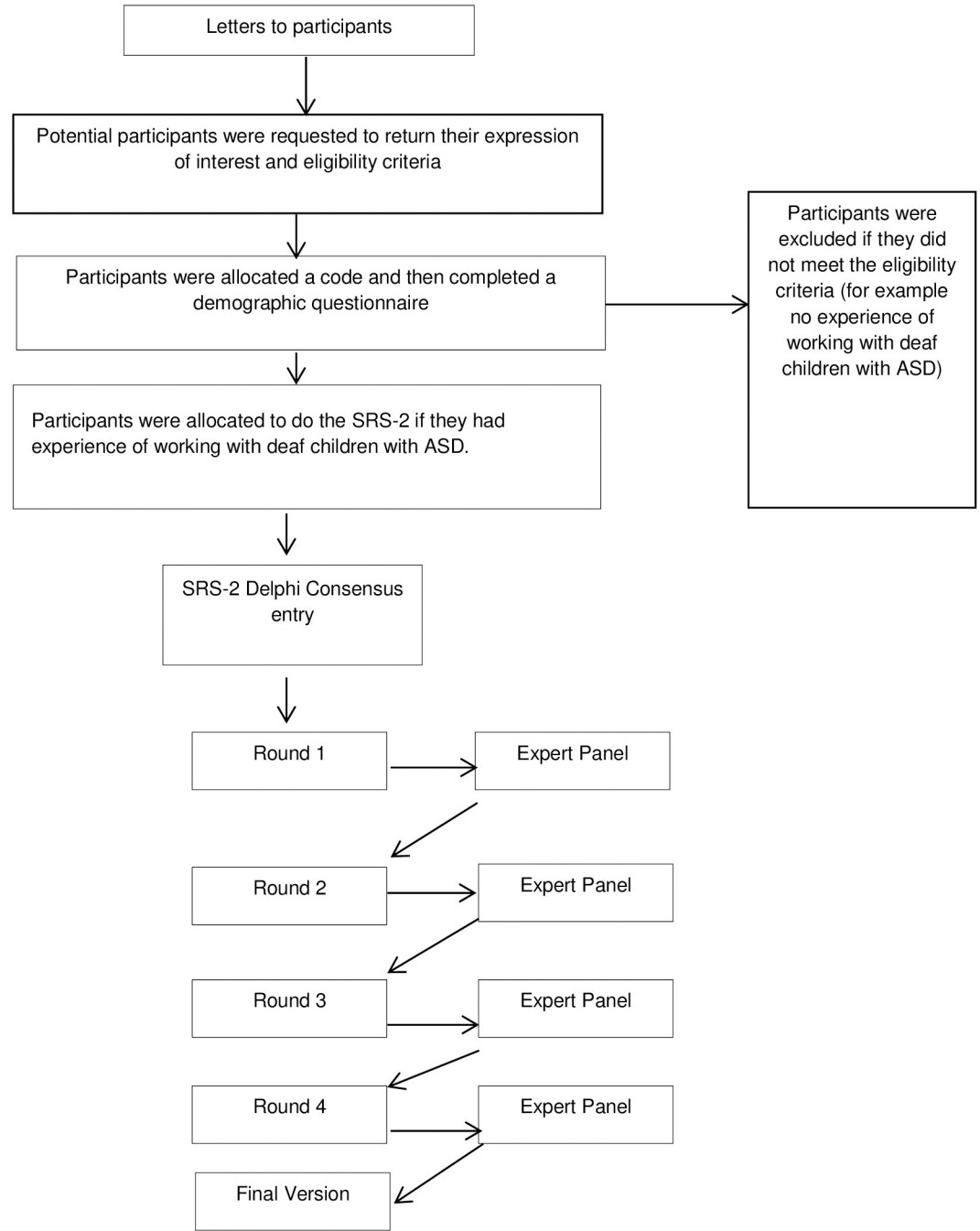

**Fig 2. Flowchart illustrating the SRS-2-Delphi International Research Panel Delphi consensus process for the SRS-2.**

where they can communicate in their main or preferred language, they may behave very differently to being with people where this is not the case. A deaf signing child in a mainstream school may be very chatty with deaf friends in sign language, but have limited engagement with hearing non-signing peers. The experts recognised that as some deaf children may socialise with deaf and/or hearing peers, it was important to provide instructions for the person

**Table 2. Participant agreement (based on the pre-established agreement thresholds described in the methodology) for individual question items during the SRS-2 rounds of the Delphi consensus.**

| Item status | Round 1 | Round 2 | Round 3 | Round 4 |
|---|---|---|---|---|
| Accepted | 16 | 25* | 10 | 8 |
| Remaining | 47 | 23 | 13 | 5 |
| Discarded | 2 | 0 | 0 | 0 |
| New item | 0 | 1* | 0 | 0 |
| Total questions | 65 | 64 | 64 | 64 |

*A new item was added at the beginning of round 2 and agreed at the end of round 2.

completing the questionnaire to consider the relevance of both the child's preferred language and the type of communications used in the social setting.

In round 2 of the Delphi Consensus process, the experts agreed to the addition of an opening statement at the beginning of the modified SRS-2. The purpose of the statement is to highlight that the respondent should answer questions with these aspects of the communication environment in mind:

"*IMPORTANT: *Deaf children will behave very differently if those around them cannot communicate with them. This questionnaire therefore needs to be filled in based on observations of the child in settings where those around them can communicate well with them, and in the child's preferred language. If your child mixes mostly with hearing children please consider this context when answering the questions. If your child mixes with deaf and hearing children please answer the questions in relation to being with deaf children*".

In addition to the opening statement, over the course of the Delphi Consensus process a number of questions were rephrased to ensure that an appropriate communicative environment was considered when scoring the items. The modes of communication used by deaf children can differ from those used by hearing children; therefore adaptations were made to ensure questions were relevant to *any* deaf child (e.g. "*Is able to communicate his or her feelings to others through words, signs or gestures*").

## The experience of being a deaf person

There are differences in lived experience between hearing and deaf children. Many deaf children have poor access to language and education and consequently may find basic linguistic structures or development of conversation challenging regardless of whether they are on the autism spectrum or not. In this context some wording modifications were proposed in order to account for the deaf lived experience.

For example, this modification takes into account the deaf person's language development:

Modified: "*Takes things more literally from other people's conversations than you would expect for his/her language development*".

Other examples included items where communication references were due to differences in deaf and hearing culture. For example, "*Touches others in an unusual way (for example may stroke a stranger's hair or their clothing)*." In deaf culture, a touch to the shoulder is an acceptable way to gain another person's attention. By contrast, in hearing culture this may not be interpreted in the same way as being appropriate to gain another person's attention.

## Structure of the instrument

The original SRS-2 was presented as two similar versions relevant for use with children of different ages (30–54 months and 4–18 years). The Delphi experts recommended combining these into one scale suitable for use with deaf children of all ages. Where the original questions differed between versions, the Delphi experts indicated which item from each version should be considered for inclusion in the SRS-2- Deaf adaptation. As discussed previously, items 52 and 53 were removed. Both versions of the SRS-2 were merged into one single version in line with the experts' recommendation. The final version was then agreed for use in the validation study.

**Creation of a new item.** During the Delphi process, the experts recommended including one additional item in order to capture theory of mind in communication. This item considers whether the child has an understanding of needing to adapt communication for others' needs:

*"Item 64: Does not appear to be aware that they might need to adapt their communication depending on whether they're communicating with a deaf or a hearing person."*

**Validation study results.** 204 deaf children and parents were eligible to take part in the study. Of these, six children were excluded because the SRS-2 Deaf adaptation was not (or only partially) completed. SRS-2 Deaf adaptation data were available for 76 deaf children with a diagnosis of ASD and 122 deaf children without an ASD diagnosis (see Fig 3). The demographic characteristics of the children by group are shown in Table 3; this includes both the

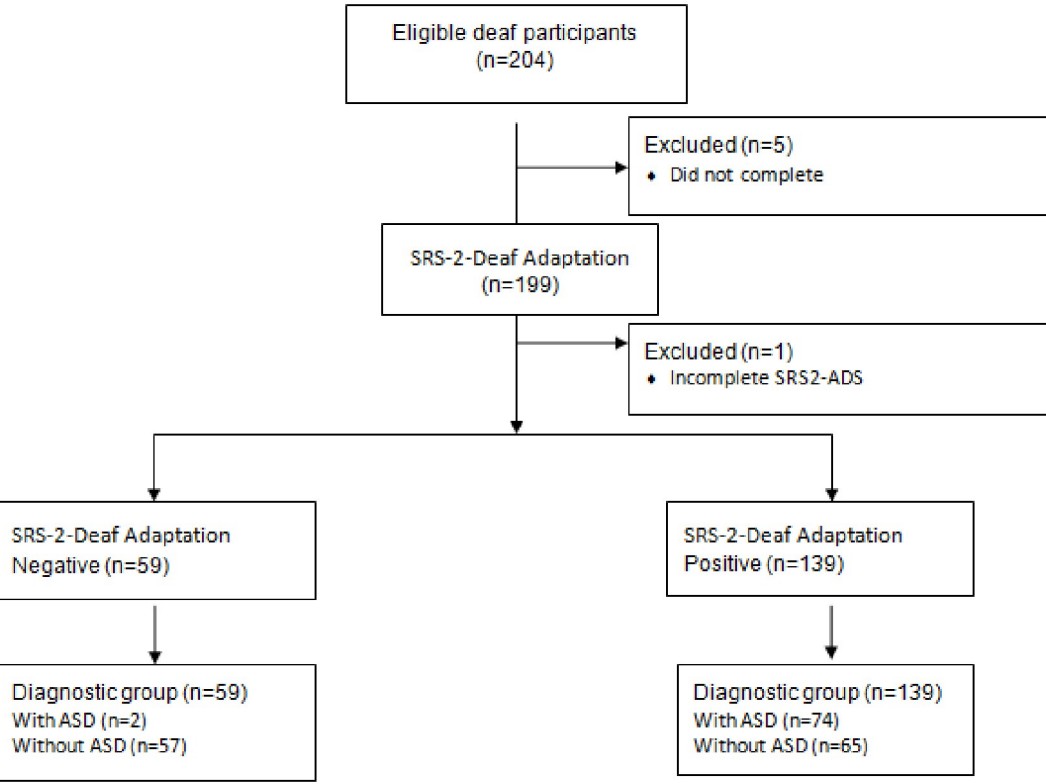

**Fig 3. STARD flowchart for SRS-2 Deaf adaptation (diagnostic group).**

**Table 3. Demographic characteristics of children and young people included in validation of the SRS-2 Deaf adaptation.**

| | | Diagnostic Group | | NICE guideline standard clinical assessments | |
| --- | --- | --- | --- | --- | --- |
| | | Deaf with ASD n = 76 | Deaf without ASD n = 122 | Deaf with ASD n = 65 | Deaf without ASD n = 104 |
| **Gender of child** | Male | 63 (83%) | 91 (75%) | 55 (85%) | 80 (77%) |
| | Female | 13 (17%) | 31 (25%) | 10 (15%) | 24 (23%) |
| **Age of child** | 0–3 | 5 (7%) | 15 (12%) | 4 (%) | 13 (%) |
| | 4–9 | 29 (38%) | 59 (48%) | 26 (%) | 52 (%) |
| | 10+ | 42 (55%) | 48 (39%) | 35 (%) | 39 (%) |
| **Ethnicity** | White | 58 (76%) | 103 (84%) | 50 (77%) | 89 (86%) |
| | Black | 2 (3%) | 5 (4%) | 1 (1%) | 4 (4%) |
| | Asian | 8 (11%) | 8 (7%) | 7 (11%) | 6 (6%) |
| | Mixed | 7 (9%) | 5 (4%) | 6 (9%) | 5 (5%) |
| | Other | 1 (1%) | 1 (1%) | 1 (2%) | 0 (0%) |
| **SRS-2-Deaf adaptation completion** | Mother | 69 (91%) | 121 (92%) | 59 (91%) | 96 (92%) |
| | Father | 1 (1%) | 7 (6%) | 1 (1%) | 5 (5%) |
| | Mother and father | 3 (4%) | 1 (1%) | 2 (3%) | 1 (1%) |
| | Other custodial adult | 3 (4%) | 2 (1%) | 3 (5%) | 2 (2%) |

'diagnostic' group and those within this group who completed the NICE guideline standard clinical assessment. All of these completed the SRS 2 Deaf adaptation. The majority of the SRS-2 Deaf adaptation questionnaires were completed by mothers (91%). We were not able to carry out any analysis of deaf parents filling in the forms as the vast majority of parents were hearing (13). This would be something for a future study.

**Validation of the SRS-2-Deaf adaptation.** There were statistically significant differences for both the raw score (p<0.001) and T-score (p<0.001) between groups. This was the case for diagnostic group and for those who received the NICE guideline standard clinical assessment (Table 4).

Using ROC curve analysis (Table 5, Fig 4), the AUC for the T-Score was 0.811 (95% CI: 0.753, 0.869) when using diagnostic group. A high sensitivity of 97% (91%-100%) was found with a cut off ≥ 60 (mild to severe) and a specificity of 44% (35%, 53%). The optimal cut-off derived from the ROC curve when using diagnostic group was ≥ 73, which gave a sensitivity of 82% (71%-90%) and a specificity of 67% (58%-75%).

Using only children who had received NICE guideline standard clinical assessments (Table 5), the AUC was 0.804 (0.740, 0.868), and with a cut off ≥ 60 the sensitivity was 97% (89%–100%) and specificity of 55% (45%, 65%). The optimal cut-off derived from the ROC

**Table 4. Summary scores for SRS-2 Deaf adaptation by diagnostic group and NICE guideline standard clinical assessments only.**

| | Deaf with ASD | | Deaf without ASD | | Mean difference (SE), 95% CI | p value |
| --- | --- | --- | --- | --- | --- | --- |
| | Mean (SD) | N | Mean (SD) | N | | |
| **Diagnostic group** | | | | | | |
| SRS-2 Deaf adaptation Total raw score | 113.7 (26.1) | 76 | 67.9 (38.7) | 122 | 45.9 (5.0), 36.0–54.9 | p<0.001 |
| SRS-2 Deaf adaptation T score | 81.0 (8.9) | 76 | 64.1 (15.3) | 122 | 16.9 (1.9), 13.1–20.7 | p<0.001 |
| **NICE guideline standard clinical assessments** | | | | | | |
| SRS-2 Deaf adaptation Total raw score | 113.1 (27.2) | 65 | 67.4 (39.1) | 104 | 45.7 (5.5), 34.8–56.7 | p<0.001 |
| SRS-2 Deaf adaptation T score | 80.6 (9.2) | 65 | 63.9 (15.6) | 104 | 16.7 (2.1), 12.5, 21.0 | p<0.001 |

**Table 5. ROC curve findings against the diagnostic group and NICE guideline standard clinical assessments.**

| | AUC (95% CI | N(Deaf with ASD/Deaf without ASD) | Cut-off | Sensitivity | Specificity |
|---|---|---|---|---|---|
| **Aspirational criteria** | | | | 80% | 70% |
| **Diagnostic group** | 0.811 (0.753, 0.869) | 198 (76/122) | 60* (Mild-Severe) | 97% (91%, 100%) | 44% (35%, 53%) |
| | | | 66* (Moderate-Severe) | 93% (85%, 98%) | 55% (46%, 64%) |
| | | | 73 (ROC Curve) | 82% (71%-90%) | 67% (58%-75%) |
| **NICE guideline standard clinical assessments** | 0.804 (0.740, 0.868) | 169 (65/104) | 60* (Mild-Severe) | 97% (89%, 100%) | 55% (45%, 65%) |
| | | | 66* (Moderate-Severe) | 92% (83%, 97%) | 53% (43%, 63%) |
| | | | 73 (ROC Curve) | 80% (68%, 89%) | 66% (56%, 75%) |

* Cut offs 60 and 66 are based on the original instrument T score threshold analysis for illustrative purposes. The T score cut off 73 is calculated from the ROC curve from the adapted version with this deaf population.

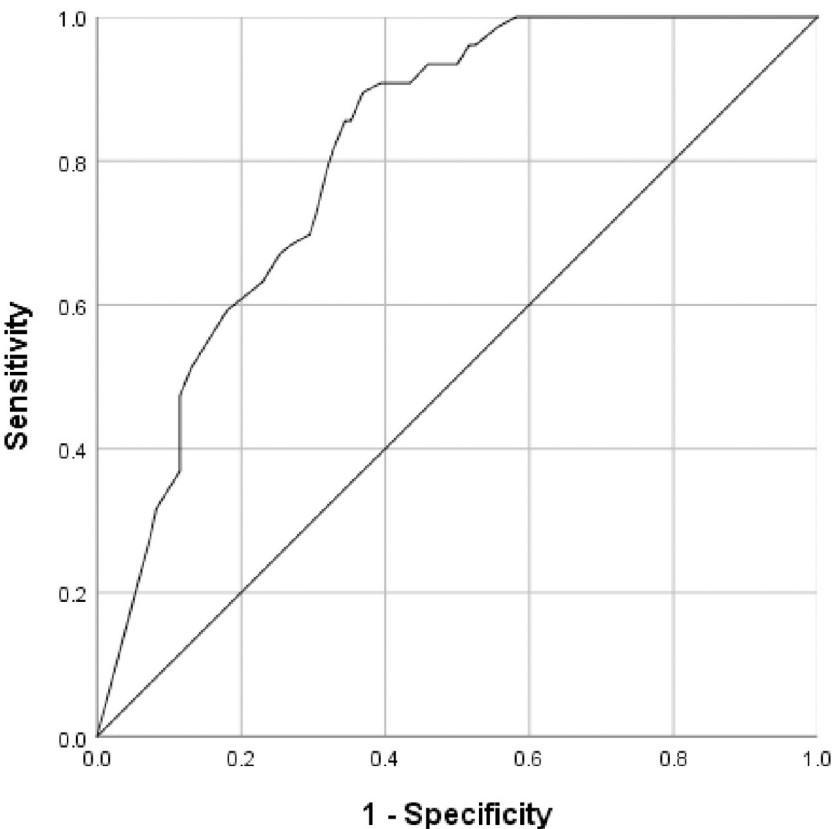

**Fig 4. ROC curve for SRS-2 Deaf adaptation T score (T1) against diagnostic group.**

**Table 6. SRS-2 Deaf adaptation new item about adapting communication against diagnostic group.**

|  | Deaf with ASD | Deaf without ASD |
|---|---|---|
| Not true | 6 (8%) | 51 (42%) |
| Sometimes true | 23 (32%) | 32 (26%) |
| Often true | 16 (22%) | 16 (13%) |
| Almost always true | 27 (38%) | 23 (19%) |
| Missing | 4 | 0 |
|  | 76 | 122 |

curve when using diagnostic group was ≥ 73, which gave a sensitivity of 80% (68%–89%) and specificity of 66% (56%–75%).

**Reliability of the SRS-2 scoring algorithm.** Cronbach's Alpha coefficient was 0.968, suggesting that the items have high internal consistency. The factor analysis revealed that the first factor included all but 12 items and accounted for 35% of the total variance. The 12 items that did not load on the primary factors are: *10. Takes things more literally from other people's conversations than you would expect for his/her language development, 25. Doesn't seem to mind being different or not on the same wavelengths as others, 49. Has one or more exceptional skills, 43. Separates easily from caregivers, 59. Is suspicious of most people, 54. Seems to react to people as if they are objects, 45. Focuses his or her attention to where others are looking, 47. Is too silly or laughs inappropriately, 55. Knows when she or he is too close to someone or is invading someone's space, 41. Wanders aimlessly from one activity to another and does not remain engaged in a particular activity, 51. Responds to clear, direct questions in ways that don't seem to make any sense or go off subject.*

The eigenvalue for the first factor was quite a bit larger than the eigenvalue for the next factor (22.582 versus 3.315). The items associated with other factors, did not reach the threshold of a factor loading of 0.5 within the factor.

The new item included in the scale "*Does not appear to be aware that they might need to adapt their communication depending on whether they're communicating with a deaf or a hearing person*" was analysed separately. As can be seen in Table 6, there was a significant difference (p<0.001) between deaf children with ASD and deaf children without ASD for this item, with children with ASD less likely to be aware that they might need to adapt their communication.

**Test/re-test of SRS-2-Deaf adaptation.** Test-retest was undertaken on 126 deaf children and the ICC was 0.897 (95%CI: 0.857, 0.927, p<0.001), hence there is evidence to support the test-retest reliability of this measurement.

## Discussion

This study has successfully designed a modified version of SRS-2 (known as the SRS-2-Deaf adaptation) which has been adapted for use with deaf children and young people across the age and ability range. The SRS-2 Deaf adaptation has been shown to successfully identify deaf children with ASD from deaf children without ASD and has good test-retest reliability.

Interestingly the deaf children diagnosed with ASD were 83% male which is remarkably close to the findings of a ten year study into the diagnosis of children in the UK (Brett et al, 2016). The validation of the SRS-2 Deaf adaptation involved some items being removed from the original SRS-2, numerous items being modified, and a new item being added, meaning that analysis was necessary to explore its performance and the need to revisit threshold scores using the original published SRS-2 scoring algorithm in the deaf population. The ROC curve

revealed that a cut-off score of ≥73 gave a sensitivity of 82%, and corresponded to a specificity of 67%. Hence, we recommend that this cut-off be used in deaf children and young people when using the SRS-2-Deaf adaptation. Encouragingly this performance is equivalent to similar instruments used for screening ASD in children/young people in the hearing population.

Changes relating to communication concentrated on both the modes of communication used by deaf children and the need to make sure that professionals scoring items consider the responses of children when they are in good communicative environments. Adaptations relating to communication also reflect the cultural differences in communication used by deaf people as well as differences in social situations where deaf children may change their behaviour or have reduced confidence than hearing children in similar social situations. The structure of the instrument includes the removal of two items and the addition of a new item, aiming to capture information about regulating language when communicating with others.

The Delphi consensus methods used in this research provided evidence of a collaborative process through increased levels of agreement as rounds progressed. This project recruited Delphi experts from a range of professions with both deaf and hearing perspectives represented. The use of online tools enabled researchers to recruit experts internationally; this was essential given the specific requirement to have experience working with deaf children/young people and ASD, and the relative rarity of expertise in both areas. International recruitment had an additional advantage of ensuring that adaptations which were culturally appropriate to use with deaf children/young people were also relevant across different countries. A limitation was that we were only able to use the English version of the SRS-2 in the Delphi process and required all experts to be able to take part using the English language. Translations of this instrument were not available in signed languages or Dutch while this research was underway and therefore all previous participant experience with the SRS-2 was assumed to be using the English language tool. Most representation came from US, UK and Australia which reflects clinical and research areas in the field and may also be a consequence of our restriction to only using the English version of the SRS-2. We were not able to find any experts in low and middle income countries in the field of ASD in deaf children. This may indicate a lack of specialist services in these countries, but it is possible we were not able to find these experts for other reasons (e.g. restriction to the English language, limited representation in the published literature or access on the internet). Improved attempts for networking may be helpful.

Those working in different time zones were not disadvantaged in regard to participation since the tools were available at any time for individuals to access. Additionally, this online approach enabled anonymity of responses. This allowed experts to express their views about the items without concerns relating to judgement from others working in different roles or with contrasting perspectives. Related to this, individual responses were independent of one another and not restricted in length which allowed each person to express their thoughts fully.

Despite the advantages of the approach taken, there were some limitations. Many of the experts struggled to find time to complete the instrument Delphi review rounds within the project timescales. Despite being able to access the Delphi consensus information at any time, this proved a barrier to completion for some people in some instances. This was exacerbated by the timing of the Independent Research Review team meetings to deliberate on Delphi participant recommendations, which needed to be pre-arranged to maximise meeting attendance, allowing limited flexibility in timescales to review the instrument. Further challenges encountered by the team included experts making comments and suggesting changes which related to the tool being used for diagnostic purposes, for which it is not intended. This was resolved during Independent Research Review team discussions and communication with Delphi International Expert Panel participants with text at the start of subsequent rounds reminding them that the SRS-2 is designed as a screening tool. Agreement was achieved by members of the

Delphi consensus and Independent Research Review team about the adapted versions of questions, and where necessary the original author was contacted on a number of occasions for further clarification or guidance (n = 5). Access to language and environments, with good communication opportunities for deaf children, were common discussions within meetings. In order to avoid the need for many additional questions, the Independent Expert Review Panel felt the most appropriate way to incorporate these two ideas was to create a statement asking respondents to complete the SRS-2-Deaf adaptation considering the child where possible when they were in an environment with access to good communication. Question changes therefore often centred on including access to different modes of language.

The translation methodology used here has been used in previous work and takes into account the specific challenges of accurate translation between, spoken/written and signed languages [38]. In particular, this takes into account the importance of involving deaf and hearing translators in both forward and backward translation groups. The methodology used demonstrates the importance of translators remaining blind to each other's outputs. It is also important for these individuals to be bilingual where possible and have an understanding of the deaf linguistic and cultural worlds. Another limitation of this work is that the screening tool has been translated into BSL but not into other international sign languages, an area that requires further work.

The validation study procedures were intended to be robust, but faced a number of challenges including limited expert clinician time and the geographical spread of the recruited deaf children with and without ASD. We also noted the varied language use and educational experiences of deaf children. This ranged from deaf children born into hearing families who used BSL as their main language, through to deaf children who live within hearing families and used mainly spoken English. School experiences were also varied with many deaf children in mainstream schools (often as the only deaf child), some in specialist deaf units in mainstream schools and some in deaf schools. Some were taught mainly in BSL while others were taught mainly orally with or without communication support. A small number of parents raised concerns initially about participating as they were worried about attracting 'incorrect' diagnoses of ASD.

Further independent replication may be helpful especially in other countries. This may need to take place, alongside existing and new work, to enable improved screening and diagnostic assessments to be made available for deaf children in other countries and in other languages.

However, it is hoped that the availability of this new adapted screening questionnaire (SRS-2-Deaf adaptation) together with the adapted versions of the ADI-R and ADOS-2 for use in the diagnostic assessments of deaf children with suspected ASD [26,27], might help to increase the confidence with which clinicians assess and diagnose ASD in deaf children and young people. Of course it is just as important (if not more important) to accurately recognise the absence of ASD as it is to diagnose it. Many deaf children have socio-emotional developmental delays and a range of other co-occurring problems that need to be identified and may require a different set of educational experiences and interventions. Placing deaf children without ASD in an ASD unit will in many cases be damaging for the child or young person's development and reduces available specialist resources for those deaf children with ASD.

Given service resource constraints, we are not recommending that all deaf children will need this suite of modified assessments but where aetiology is uncertain or where there is complexity we believe that these measures specifically modified for use with deaf children/young people, give a structure and clarity to the assessment process that will be helpful. This new screening tool will in our opinion be a helpful first stage in the process.

We hope that our work will improve awareness of ASD in deaf children and also the importance of careful diagnostic assessment in this group.

## Conclusion

A single version of the SRS-2 Deaf adaptation has been created for use with deaf children and young people. This adapted screening questionnaire has also been translated into British Sign Language to enable access to this questionnaire for deaf parents/carers whose preferred language is BSL. However in practice there were only three participants who used the BSL translation. This would not give sufficient reliability for this sub-group, which would need to be the subject of a specific future study. This research suggests that the SRS-2 Deaf adaptation is a valid screening tool for use with deaf children who may require screening for an Autistic Spectrum Disorders and that this measure has psychometric properties equivalent to similar instruments used for screening ASD in children/young people in the hearing population.

It is hoped that with the use of improved ASD screening and diagnostic assessment processes may contribute to reducing inequities in diagnostic age and accuracy in deaf children. This in turn should facilitate better planning of appropriate educational provision and parental support earlier with the aim of improving both the clinical experiences and outcomes for deaf children/young people and their families.

## Supporting information

**S1 Appendix.**
(DOCX)

## Acknowledgments

The team would like to thank all of the clinicians and parents who gave their expertise as part of the Delphi consensus process. Additional thanks to Richard Ogden, Hannah George, Kate Rowley, Jen Dodd, Rachael Hayes and Helen McConnachie for advice. Thanks also to John Constantino the author of the original SRS-2 and the Western Psychological Services for their support of this work. Thanks to Amelia Taylor for assisting with formatting and referencing of the paper.

## Author Contributions

**Conceptualization:** Barry Wright.

**Data curation:** Helen Phillips.

**Formal analysis:** Martin Bland, Victoria Allgar.

**Funding acquisition:** Barry Wright.

**Investigation:** Barry Wright, Helen Phillips, Jennifer Sweetman, Rachel Hodkinson, Amelia Ralph-Lewis, Emily Hayward, Alice Brennan, Josie Mulloy, Natalie Day.

**Methodology:** Barry Wright, Ann Le Couteur.

**Project administration:** Barry Wright, Helen Phillips, Ann Le Couteur.

**Resources:** Barry Wright, Ann Le Couteur.

**Supervision:** Barry Wright.

**Validation:** Barry Wright, Ann Le Couteur, Jennifer Sweetman, Rachel Hodkinson, Emily Hayward, Alice Brennan, Josie Mulloy, Natalie Day, Victoria Allgar.

**Visualization:** Barry Wright, Helen Phillips, Jennifer Sweetman, Victoria Allgar.

**Writing – original draft:** Barry Wright, Helen Phillips.

**Writing – review & editing:** Barry Wright, Helen Phillips, Ann Le Couteur, Jennifer Sweetman, Rachel Hodkinson, Emily Hayward, Alice Brennan, Josie Mulloy, Natalie Day, Martin Bland, Victoria Allgar.

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
