## [Decision Letter · Decision Letter 0]

20 Aug 2020

PONE-D-20-16637

Modifying and Validating the Social Responsiveness Scale Edition 2 for use with Deaf Children and Young People

PLOS ONE

Dear Dr. Wright,

Thank you for submitting your manuscript to PLOS ONE. After careful consideration, we feel that it has merit but does not fully meet PLOS ONE’s publication criteria as it currently stands. Therefore, we invite you to submit a revised version of the manuscript that addresses the points raised during the review process.

We look forward to receiving your revised manuscript.

Kind regards,

Francesca Chiesi

Academic Editor

PLOS ONE

Journal Requirements:

2. You indicated that you had ethical approval for your study. In your Methods section, please ensure you have also stated whether you obtained informed consent from parents or guardians of the minors included in the study or whether the research ethics committee or IRB specifically waived the need for their consent. I you obtained consent, please ensure that you have specified what type you obtained (for instance, written or verbal, and if verbal, how it was documented and witnessed).

3. Please ensure that you refer to Figure 3 in your text as, if accepted, production will need this reference to link the reader to the figure.

4. We note you have included a table to which you do not refer in the text of your manuscript. Please ensure that you refer to Table 2 in your text; if accepted, production will need this reference to link the reader to the Table.

Reviewers' comments:

Reviewer's Responses to Questions

**Comments to the Author**

1. Is the manuscript technically sound, and do the data support the conclusions?

Reviewer #1: Yes

Reviewer #2: Yes

Reviewer #3: Yes

2. Has the statistical analysis been performed appropriately and rigorously? 

Reviewer #1: Yes

Reviewer #2: Yes

Reviewer #3: Yes

3. Have the authors made all data underlying the findings in their manuscript fully available?

Reviewer #1: Yes

Reviewer #2: No

Reviewer #3: Yes

4. Is the manuscript presented in an intelligible fashion and written in standard English?

Reviewer #1: Yes

Reviewer #2: Yes

Reviewer #3: No

5. Review Comments to the Author

Reviewer #1: Review of “Modifying and Validating the Social Responsiveness Scale Edition 2 for use with Deaf Children and Young People”

This paper describes a recent adaptation and validation of the Social Responsiveness Scale (SRS-2) for use with deaf children. There are currently no validated instruments available for the screening or diagnosis of autism spectrum disorder (ASD) in deaf children. As such, this endeavor represents an essential step forward in beginning to provide culturally and linguistically appropriate tools for deaf children. I applaud the researchers for taking on this work, as it is sorely needed.

Overall the paper is quite complete, well written, and the procedure is adequately described. It is easy to understand how the researchers went about the adaptation and validation procedure, and the procedures selected are in line with best practices. The authors also successfully show that the tool is able to discriminate between ASD and non-ASD children (as assigned by clinical criteria). The sensitivity of the instrument is very good, only missing 2 children with ASD. Specificity also appears to be acceptable, despite initially classifying many non-ASD children as ASD-positive.

My concerns about the paper are mostly relatively minor. However, the paper could be strengthened by addressing the following issues:

1) Throughout the paper there is confusion about whether or not the instrument used in the validation study was in English or in British Sign Language (BSL). I understand that the SRS-2 was adapted in written English and then translated and again back-translated into BSL. However it is unclear how many families who then completed the instrument did so in English or BSL, and if the sensitivity and specificity are comparable in each modality. There are a number of places in the manuscript where this should be clarified. For example, p. 7, does “this newly modified instrument” refer to the BSL instrument, or Deaf-adapted English instrument? Similarly, p. 27, Table 3, including information about the hearing status of the parents and modality that the SRS was completed in would help the reader to better understand how the SRS was completed and the nature of the sample. There is also no discussion of any translation issues encountered in this process, although the translation from English into BSL is no small undertaking. If possible, I would like to see more detail about this.

2) The way that the paper is structured, it appears that some of the results are hinted at in the method. For example, p. 9, “some questions were deemed unsuitable to assess ASD in deaf children and were discarded” makes the reader want to know which questions were discarded, but those are not reported until the Results. Authors should consider restructuring this paragraph so as to not beg these questions. A construction such as “If questions were deemed unsuitable to assess ASD in deaf children, they were discarded” may work better. This is true for other sections in the Method: e.g., when you describe the Independent Research Review Team on p. 11, I was left wondering how many participants from each group were included.

3) p. 25, text reads: “Many deaf children have poor access to language and education and consequently, may not have a good understanding of English metaphor. This is particularly prevalent [sic] if they are exposed to spoken and signed languages, each having very different use of metaphorical expressions.” I find this section a bit problematic. Having poor access to language leads to much bigger problems than merely not understanding metaphors – it can lead to not understanding basic linguistic structures! Additionally, being exposed to sign and speech does not seem to me to be the problem – if a child is exposed to both, then they would likely understand metaphor in both modalities. However if a child is only exposed to speech, and has poor access to spoken language, they are unlikely to understand metaphor in either modality. In other words, the effects of language deprivation are not clearly described here, and currently the text is a bit misleading. In addition, the item that this section introduces does not focus on metaphor, but rather on general pragmatic abilities (“takes things too literally and doesn’t get the real meaning of a conversation”). I suggest that this section be rewritten to be clarified.

Global terminology concerns:

a) I’m not sure that the term “subjects” is correct when referring to deaf children (e.g. on p. 3 in the introduction) since the children are not technically participants in research. Similarly, the use of the term “participants” to refer to the children (e.g. in Fig. 3) is confusing, since both parents and children are technically participants in the research. I would suggest clarifying by using terms such as “children” and “parents” to specify which part of the research you are referring to.

b) please use “ASD” rather than “autism” throughout the paper, line with current DSM terminology.

Other/minor comments:

p. 3: definition of the term “deaf” is circular – please define in terms of hearing loss

p. 4: please report the comparable statistics of age of diagnosis for hearing children so that the reader can see the discrepancy between deaf and hearing

Missing punctuation: p. 5, end of paragraph 1; also, p. 25, line 562: punctuation missing from end of sentence. Also please use the serial (Oxford) comma between elements in a series of three of more items, in line with APA style.

p. 6, lines 128-131: this sentence is a bit awkward and confusing. The way it is currently written it reads as though the SRS is a parent semi-structured interview rather than the ADI-R

p. 8, there are two places where “see below” appears in the text; it would be better to replace these with more concrete directions (i.e., “see Fig.1”, etc.).

The Delphi Consensus Method is referred to rather extensively before it is finally described on p. 8. In addition, the reliability and validity of this method is described on lines 194-199, but should be described earlier.

The International Expert Panel consists of experts from a variety of countries, including non-English speaking countries, which introduces a wrinkle into the methodology, though not a big one (nearly all of the respondents were from English speaking countries). I find the description of the Panel on p. 9 a bit lacking. Perhaps you could mention here that most of the respondents were from English speaking countries, not just in Table 1.

p. 15, use of the SCQ also warrants caution, for the same reasons that the SRS-2 is being adapted. It is not validated for deaf children and contains items that are inappropriate (e.g., does the child respond when name is called). This should perhaps be noted.

Table 2: Round 1 says that 47 questions remained, but in Round 2 the sum of accepted and remaining items is 48. Is the reason for this discrepancy the addition of the new item? Perhaps I would asterisk the accepted item number as well to make this clear, if this is the case.

Table 3, p. 27, the column headers (“Diagnostic Group” vs “NICE guideline standard clinical assessments”) are confusing. Is one group a subset of the other? If so, perhaps this should be reformatted. Currently it looks like they are two independent groups.

p. 29, please specify in the text which groups there are statistically significant differences between.

p. 36, line 759: “…required all experts to be able to take part using the English language.” How did the deaf experts take part? Using written English?

Will the Appendix be included in the publication?

Reviewer #2: This well-conceived and soundly-executed study describes the modification and validation of a screening instrument designed to identify deaf children who should receive a full diagnostic evaluation for autism spectrum disorder. An existing screener was adapted via the guidance of a panel of international (English-speaking) experts, and was validated with a sample of 198 deaf children who did or did not already have an ASD diagnosis: a very large sample for this field. The modified screener had excellent internal consistency and test-retest reliability. Sensitivity was good and specificity was adequate for a screener.

I have no major concerns about this study, and only a handful of minor questions.

1. I am curious as to whether the recruitment efforts for the validation study specifically included ASD communities who view their diagnosis as a dimension of human neurodiversity rather than as pathology. (I am not sufficiently familiar with the British organizations listed to determine this on my own.) If so, I would view that as a strength and recommend that the authors make that explicit. If not, I would view that as a limitation and recommend that the authors include that in the list of limitations provided in the discussion.

2. Given the staggering heterogeneity of DHH children’s backgrounds, I would appreciate more detail on what constituted “a comprehensive history of the child’s development, family, and medical history”, especially with regard to their language history. Perhaps this information (or, at the very least, the questionnaires used) could be included as supplemental material.

3. I find Table 2 difficult to interpret, and note that it is not discussed anywhere in the text.

4. p. 24, ln 538-544: I support the inclusion of a caveat message like this; however, the particular verbiage here puzzles me. For a child who mixes mostly with hearing children, the caveat is barely needed, since those children would presumably be the only context that the carers could consider. For a child who mixes with both deaf and hearing children, the direction to consider only interactions with deaf children strikes me as inappropriate. The assumption seems to be that socializing with deaf children with allow barrier-free communication, but that is simply not the case. A deaf child who is a fluent signer may be surrounded by deaf children whose language skills are far below their developmental age. A deaf child who has been raised with a focus on listening and spoken language may be placed in a signing environment for the first time and be surrounded by deaf children who are using a language that is entirely new to them. It would seem more appropriate to simply truncate the instruction after the second sentence.

p. 25 ln 572-273: I assumed that the example item given here belonged to the original set; I recommend simply removing the text of the item from this sentence.

Table 3: I would appreciate more comment about the extent to which the demographics of the sample do or do not reflect the demographics of the UK population, as well as some discussion about the potential for existing assessments to over-/under-identify ASD in various demographic groups. Is there reason to believe that the SRS-2 Deaf Adaption fares any better or worse on this front?

Reviewer #3: Thank you for the opportunity to review this interesting manuscript. The authors report on a modification process and validation study of the SRS-2 for use in screening children and youth who are deaf for ASD. This is an area in which improvements are definitely needed, due to the complexities of assessing children who are deaf given differences in language development, behaviors in context, etc. Following a Delphi method informing modification of items, several psychometric properties of the adapted tool were assessed with a sample of British children (N=198) and parents; approximately 38% of the children met diagnostic criteria for ASD. ROC analyses were used to identify an optimal cut-point on the adapted tool. Strengths include the use of the Delphi process to engage international experts to review the original SRS-2 items, as well as subsequently modified items, over 4 waves of deliberation to develop an adapted version of the instrument. Participants for the validation study appear to have been recruited from a broad array of settings, increasing generalizability. The psychometric assessments of the screening tool, according to the authors, suggested that the adapted tool performs with similar levels of sensitivity and specificity as other ASD screening tools with hearing populations. The focus on this understudied population is important and appreciated.

Some weaknesses of the manuscript in its current form were noted, and these are listed below.

1) Table 2 and the text immediately following are very difficult to follow in terms of the process of reviewing, accepting, modifying, and discarding items. Perhaps the authors could describe this process in a different way. Just one example of difficulties interpreting the table – at the end of round 4, there were 5 “remaining” items, and it is unclear what “remained” means – they ended up being included in the final screening tool, but why?

2) Rather than the piecemeal description of which items were modified and why, it would be very helpful if the authors could organize this information in a more meaningful way. For example – if there were a certain number of “common” rationales for modifying items, perhaps a table could be developed listing each item, whether it was changed or not, and the rationale for the change. The final tool is presented in the additional materials, and it could instead be a part of the manuscript reporting the modifications, deletions, and addition to the items of the original SRS-2.

3) An example of a rationale for modifying an item that was not very clear is re: the item that was replaced because of the phrase “too tense,” with a statement that it was not clear how being “too tense” could be evaluated consistently in children who are deaf. That example begs the question of how being “too tense” is evaluated in hearing children. The rationale is not terribly clear – it may just be a vague item that does not clearly measure what it is intended to measure, regardless of the population of children targeted.

4) Similarly, additional information and justification for the items pulled from the two age-based versions of the SRS-2 would be helpful (i.e., not just that the group decided to go with an item from one rather than the other, but why).

5) Information on which 12 items did not load on the primary factor is needed.

6) It is unclear why parents of hearing children were not included in the test-retest reliability assessment.

7) Table 3 – percentages are missing for the child age categories under the NICE column.

8) Line 665 says diagnostic group, should refer to the NICE assessment group

9) Line 737, please clarify that 73% refers to the t-score

10) Line 751 refers to increasing agreement among the expert panel, but the authors initially described lowering the bar for agreement from 80% to 75% percent for the fourth round – please clarify.

11) As addressed in the introduction section, 90% of parents of children who are deaf are hearing, themselves. Were there deaf parents/carers in the study? There is no mention of this in Table 3. If not, the sign language version of the screening tool was not actually assessed for reliability and validity (and even if there were a few deaf parents/carers, there were likely not enough to make conclusions re: the translated tool’s psychometrics – this would need to be addressed in future studies).

12) I would strongly recommend a very close copy-editing of the manuscript. There are many issues with punctuation and phrasing that detract from the importance of the work.

6. PLOS authors have the option to publish the peer review history of their article (what does this mean?). If published, this will include your full peer review and any attached files.

Reviewer #1: No

Reviewer #2: No

Reviewer #3: No

---

## [Author Response · Author response to Decision Letter 0]

14 Oct 2020

1. Is the manuscript technically sound, and do the data support the conclusions?

 Reviewer #1: Yes

Reviewer #2: Yes

Reviewer #3: Yes

2. Has the statistical analysis been performed appropriately and rigorously? 

 Reviewer #1: Yes

Reviewer #2: Yes

Reviewer #3: Yes

3. Have the authors made all data underlying the findings in their manuscript fully available?

 Reviewer #1: Yes

Reviewer #2: No

Reviewer #3: Yes

 We have addressed this

4. Is the manuscript presented in an intelligible fashion and written in standard English?

 Reviewer #1: Yes

Reviewer #2: Yes

Reviewer #3: No

 We have addressed this

Reviewer 1

This paper describes a recent adaptation and validation of the Social Responsiveness Scale (SRS-2) for use with deaf children. There are currently no validated instruments available for the screening or diagnosis of autism spectrum disorder (ASD) in deaf children. As such, this endeavor represents an essential step forward in beginning to provide culturally and linguistically appropriate tools for deaf children. I applaud the researchers for taking on this work, as it is sorely needed.

Overall the paper is quite complete, well written, and the procedure is adequately described. It is easy to understand how the researchers went about the adaptation and validation procedure, and the procedures selected are in line with best practices. The authors also successfully show that the tool is able to discriminate between ASD and non-ASD children (as assigned by clinical criteria). The sensitivity of the instrument is very good, only missing 2 children with ASD. Specificity also appears to be acceptable, despite initially classifying many non-ASD children as ASD-positive.

My concerns about the paper are mostly relatively minor. However, the paper could be strengthened by addressing the following issues:

Comment No. 

Comments 

Response

1 

Throughout the paper there is confusion about whether or not the instrument used in the validation study was in English or in British Sign Language (BSL). I understand that the SRS-2 was adapted in written English and then translated and again back-translated into BSL. However it is unclear how many families who then completed the instrument did so in English or BSL, and if the sensitivity and specificity are comparable in each modality. There are a number of places in the manuscript where this should be clarified. For example, p. 7, does “this newly modified instrument” refer to the BSL instrument, or Deaf-adapted English instrument? Similarly, p. 27, Table 3, including information about the hearing status of the parents and modality that the SRS was completed in would help the reader to better understand how the SRS was completed and the nature of the sample. There is also no discussion of any translation issues encountered in this process, although the translation from English into BSL is no small undertaking. If possible, I would like to see more detail about this.

 As suggested (Pg.7) we have made it clear that the SRS-2-DA written English is validated and not the translated BSL version. 

The specificity and sensitivity only relates to the SRS-2-DA written English version.

The vast majority of the SRS-2 DA were completed by hearing parents and we were not able to analyse deaf parents separately because of low numbers. This is in line with the fact that 93% of deaf children are born to hearing parents. We have added this information.

2 

The way that the paper is structured, it appears that some of the results are hinted at in the method. For example, p. 9, “some questions were deemed unsuitable to assess ASD in deaf children and were discarded” makes the reader want to know which questions were discarded, but those are not reported until the Results. Authors should consider restructuring this paragraph so as to not beg these questions. A construction such as “If questions were deemed unsuitable to assess ASD in deaf children, they were discarded” may work better. This is true for other sections in the Method: e.g., when you describe the Independent Research Review Team on p. 11, I was left wondering how many participants from each group were included.

 Many thanks for these comments. We have stayed with the convention of keeping the results in the results section rather than letting them appear in the methods section.

3 

p. 25, text reads: “Many deaf children have poor access to language and education and consequently, may not have a good understanding of English metaphor. This is particularly prevalent [sic] if they are exposed to spoken and signed languages, each having very different use of metaphorical expressions.” I find this section a bit problematic. Having poor access to language leads to much bigger problems than merely not understanding metaphors – it can lead to not understanding basic linguistic structures! Additionally, being exposed to sign and speech does not seem to me to be the problem – if a child is exposed to both, then they would likely understand metaphor in both modalities. However if a child is only exposed to speech, and has poor access to spoken language, they are unlikely to understand metaphor in either modality. In other words, the effects of language deprivation are not clearly described here, and currently the text is a bit misleading. In addition, the item that this section introduces does not focus on metaphor, but rather on general pragmatic abilities (“takes things too literally and doesn’t get the real meaning of a conversation”). I suggest that this section be rewritten to be clarified.

 We agree with these comments and have made changes accordingly to better explain this issue.

4 

Global terminology concerns:

a) I’m not sure that the term “subjects” is correct when referring to deaf children (e.g. on p. 3 in the introduction) since the children are not technically participants in research. Similarly, the use of the term “participants” to refer to the children (e.g. in Fig. 3) is confusing, since both parents and children are technically participants in the research. I would suggest clarifying by using terms such as “children” and “parents” to specify which part of the research you are referring to.

b) please use “ASD” rather than “autism” throughout the paper, line with current DSM terminology.

 We have changed the terms ‘subjects’ and ‘participants’ in line with the reviewers suggestions.

We have also used the term ASD where this is more appropriate than autism. 

Other/minor comments:

5 p. 3: definition of the term “deaf” is circular – please define in terms of hearing loss The definition of deaf here is in audiological terms since mild, moderate, severe or profoundly deaf are audiologically defined and this is referenced. The additional phrase about communication modality is to highlight the fact that we did not exclude people with particular communication modalities (e.g. use of sign language)

6 p. 4: please report the comparable statistics of age of diagnosis for hearing children so that the reader can see the discrepancy between deaf and hearing The figures mention deaf children and also children in general (it is not possible to obtain figures from general population studies where deaf children have been excluded from the children in general group as this subgrouping has not been undertaken in any published studies)

7 Missing punctuation: p. 5, end of paragraph 1; also, p. 25, line 562: punctuation missing from end of sentence. Also please use the serial (Oxford) comma between elements in a series of three of more items, in line with APA style. Thank you. We have added these missing punctuation marks.

8 p. 6, lines 128-131: this sentence is a bit awkward and confusing. The way it is currently written it reads as though the SRS is a parent semi-structured interview rather than the ADI-R We have rewritten this sentence to make it clearer.

9 p. 8, there are two places where “see below” appears in the text; it would be better to replace these with more concrete directions (i.e., “see Fig.1”, etc.). More concrete signposting has now been added to these sections.

10 The Delphi Consensus Method is referred to rather extensively before it is finally described on p. 8. In addition, the reliability and validity of this method is described on lines 194-199, but should be described earlier. This section has now been restructured in line with reviewer comments.

11 The International Expert Panel consists of experts from a variety of countries, including non-English speaking countries, which introduces a wrinkle into the methodology, though not a big one (nearly all of the respondents were from English speaking countries). I find the description of the Panel on p. 9 a bit lacking. Perhaps you could mention here that most of the respondents were from English speaking countries, not just in Table 1. In keeping with the standard structure of methods and results sections, additional information has been added to the text within the results section to provide this additional detail.

12 p. 15, use of the SCQ also warrants caution, for the same reasons that the SRS-2 is being adapted. It is not validated for deaf children and contains items that are inappropriate (e.g., does the child respond when name is called). This should perhaps be noted.

 Additional detail has now been added to highlight this issue.

13 Table 2: Round 1 says that 47 questions remained, but in Round 2 the sum of accepted and remaining items is 48. Is the reason for this discrepancy the addition of the new item? Perhaps I would asterisk the accepted item number as well to make this clear, if this is the case.

 Thank you. As suggested, an additional asterisk has now been added to clarify this discrepancy.

14 Table 3, p. 27, the column headers (“Diagnostic Group” vs “NICE guideline standard clinical assessments”) are confusing. Is one group a subset of the other? If so, perhaps this should be reformatted. Currently it looks like they are two independent groups.

 Additional information has been added to the paragraph before the table to add clarification about the 2 different groups.Line:630

15 p. 29, please specify in the text which groups there are statistically significant differences between.

 In line with CONSORT, the descriptive characteristics are presented but the groups are not compared statistically.

16 p. 36, line 759: “…required all experts to be able to take part using the English language.” How did the deaf experts take part? Using written English?

 This instrument was not available in a Dutch format at the time this research was conducted. The participants had previous experience using the English language tool. Additional information has been added to the text to reflect this.

17 

Will the Appendix be included in the publication?

 There will be no appendix with the questionnaire as this is now licenced by WPS.

Reviewer 2

This well-conceived and soundly-executed study describes the modification and validation of a screening instrument designed to identify deaf children who should receive a full diagnostic evaluation for autism spectrum disorder. An existing screener was adapted via the guidance of a panel of international (English-speaking) experts, and was validated with a sample of 198 deaf children who did or did not already have an ASD diagnosis: a very large sample for this field. The modified screener had excellent internal consistency and test-retest reliability. Sensitivity was good and specificity was adequate for a screener.

I have no major concerns about this study, and only a handful of minor questions.

Comment No. 

Comments 

Response

1

I am curious as to whether the recruitment efforts for the validation study specifically included ASD communities who view their diagnosis as a dimension of human neurodiversity rather than as pathology. (I am not sufficiently familiar with the British organizations listed to determine this on my own.) If so, I would view that as a strength and recommend that the authors make that explicit. If not, I would view that as a limitation and recommend that the authors include that in the list of limitations provided in the discussion.

 We included all families and people with ASD including those who view ASD in a positive light. Indeed the lead author has written a paper about this [Wright, B., Spikins, P., & Pearson, H. (2020). Should Autism Spectrum Conditions Be Characterised in a More Positive Way in Our Modern World?] We have added this to the text.

2 

Given the staggering heterogeneity of DHH children’s backgrounds, I would appreciate more detail on what constituted “a comprehensive history of the child’s development, family, and medical history”, especially with regard to their language history. Perhaps this information (or, at the very least, the questionnaires used) could be included as supplemental material.

 The ADI-R Deaf adaptation included material about the causes of being deaf, medical and cultural history, linguistic and communication history and choices, their family make up and other deaf people in the family, family communication history and profile. This is in the ADI-R Deaf adaptation which is now available from WPS online and we have therefore not included it as supplementary material here.

3 

I find Table 2 difficult to interpret, and note that it is not discussed anywhere in the text.

 Thank you. This is an oversight and a reference to Table 2 has been added which accompanies a more detailed description of the decisions made about items during the Delphi process.

4 

p. 24, ln 538-544: I support the inclusion of a caveat message like this; however, the particular verbiage here puzzles me. For a child who mixes mostly with hearing children, the caveat is barely needed, since those children would presumably be the only context that the carers could consider. For a child who mixes with both deaf and hearing children, the direction to consider only interactions with deaf children strikes me as inappropriate. The assumption seems to be that socializing with deaf children with allow barrier-free communication, but that is simply not the case. A deaf child who is a fluent signer may be surrounded by deaf children whose language skills are far below their developmental age. A deaf child who has been raised with a focus on listening and spoken language may be placed in a signing environment for the first time and be surrounded by deaf children who are using a language that is entirely new to them. It would seem more appropriate to simply truncate the instruction after the second sentence.

 This was the subject of careful discussion in the expert Delphi Consensus panel. In relation to the agreed final consensus about this we agree with the reviewer that the task of the person filling in the questionnaire is to make sure that they fill in the questionnaire based on contexts where the child is in communication conditions (optimum for them) that allows a fair assessment. For example it would be unfair to fill in the questionnaire based on observations of a child with spoken English for the first time in a school where Cantonese is being spoken. We have checked the text and believe that this comes across.

5 

p. 25 ln 572-273: I assumed that the example item given here belonged to the original set; I recommend simply removing the text of the item from this sentence.

 Text examples from the original items have now been removed from this section in line with this suggestion.

6 

Table 3: I would appreciate more comment about the extent to which the demographics of the sample do or do not reflect the demographics of the UK population, as well as some discussion about the potential for existing assessments to over-/under-identify ASD in various demographic groups. Is there reason to believe that the SRS-2 Deaf Adaption fares any better or worse on this front?

 We have added to the discussion comparison between table 3 results and the most recent UK 10 year study.

We are not able to answer the second question of the reviewer in this section from our study results, although it is of course an important question for future study.

Reviewer 3

Thank you for the opportunity to review this interesting manuscript. The authors report on a modification process and validation study of the SRS-2 for use in screening children and youth who are deaf for ASD. This is an area in which improvements are definitely needed, due to the complexities of assessing children who are deaf given differences in language development, behaviors in context, etc. Following a Delphi method informing modification of items, several psychometric properties of the adapted tool were assessed with a sample of British children (N=198) and parents; approximately 38% of the children met diagnostic criteria for ASD. ROC analyses were used to identify an optimal cut-point on the adapted tool. Strengths include the use of the Delphi process to engage international experts to review the original SRS-2 items, as well as subsequently modified items, over 4 waves of deliberation to develop an adapted version of the instrument. Participants for the validation study appear to have been recruited from a broad array of settings, increasing generalizability. The psychometric assessments of the screening tool, according to the authors, suggested that the adapted tool performs with similar levels of sensitivity and specificity as other ASD screening tools with hearing populations. The focus on this understudied population is important and appreciated.

Some weaknesses of the manuscript in its current form were noted, and these are listed below.

Comment No. 

Comments 

Response

1 

Table 2 and the text immediately following are very difficult to follow in terms of the process of reviewing, accepting, modifying, and discarding items. Perhaps the authors could describe this process in a different way. Just one example of difficulties interpreting the table – at the end of round 4, there were 5 “remaining” items, and it is unclear what “remained” means – they ended up being included in the final screening tool, but why?

 Thank you for your comments on this section. The text has been altered to offer clarity around this phase of work.

2 Rather than the piecemeal description of which items were modified and why, it would be very helpful if the authors could organize this information in a more meaningful way. For example – if there were a certain number of “common” rationales for modifying items, perhaps a table could be developed listing each item, whether it was changed or not, and the rationale for the change. The final tool is presented in the additional materials, and it could instead be a part of the manuscript reporting the modifications, deletions, and addition to the items of the original SRS-2.

 We have removed the final tool from the ‘additional materials’ as it is now licenced by WPS .

The modifications are organised into subgroups namely changes as a result of : 

The communication environment

The experience of being a deaf person

The structure of the instrument and the 

Creation of new items based on research

These were categories suggested by our PPI group and for this reason we would like to stick to this organisation of material.

3 An example of a rationale for modifying an item that was not very clear is re: the item that was replaced because of the phrase “too tense,” with a statement that it was not clear how being “too tense” could be evaluated consistently in children who are deaf. That example begs the question of how being “too tense” is evaluated in hearing children. The rationale is not terribly clear – it may just be a vague item that does not clearly measure what it is intended to measure, regardless of the population of children targeted.

 We agree that this example could be interpreted in the same way for hearing children and have therefore removed it.

4 Similarly, additional information and justification for the items pulled from the two age-based versions of the SRS-2 would be helpful (i.e., not just that the group decided to go with an item from one rather than the other, but why).

 We have added an explanation for this as requested.

5 Information on which 12 items did not load on the primary factor is needed.

 We have added the details of these 12 different items on line: 767 to make it clearer for readers.

6

 It is unclear why parents of hearing children were not included in the test-retest reliability assessment. This is because we wanted specifically to check the test-retest reliability of the Deaf adaptation used for deaf children. The hearing children were included to compare symptomatology profiles. We have made that clear in the methods.

7 Table 3 – percentages are missing for the child age categories under the NICE column.

 There is no missing data

8 Line 665 says diagnostic group, should refer to the NICE assessment group

 Text updated

This was the case for diagnostic group and in the sensitivity analysis using the NICE guideline standard clinical assessment (Table 4).

9

 Line 737, please clarify that 73% refers to the t-score

 Added to footnotes

10 Line 751 refers to increasing agreement among the expert panel, but the authors initially described lowering the bar for agreement from 80% to 75% percent for the fourth round – please clarify.

 The agreement of participants agreed as round progressed as some items were adapted. The 80/75% bar change is standard in Delphi processes to make sure that agreement is reached and was prospectively set. It is mentioned that this is pre-specified in the methods section (line 282).

11 As addressed in the introduction section, 90% of parents of children who are deaf are hearing, themselves. Were there deaf parents/carers in the study? There is no mention of this in Table 3. If not, the sign language version of the screening tool was not actually assessed for reliability and validity (and even if there were a few deaf parents/carers, there were likely not enough to make conclusions re: the translated tool’s psychometrics – this would need to be addressed in future studies).

 We checked our recruitment information and only 3 participants used the BSL video for the SRS-2, therefore we are not powered for reliability in this sub-group as numbers are not large enough.

We have added this into the conclusion.

12 I would strongly recommend a very close copy-editing of the manuscript. There are many issues with punctuation and phrasing that detract from the importance of the work.

 We have had the paper proof read.

---

## [Editor Report · Decision Letter 1]

17 Nov 2020

Modifying and Validating the Social Responsiveness Scale Edition 2 for use with Deaf Children and Young People

PONE-D-20-16637R1

Dear Dr. Wright,

We’re pleased to inform you that your manuscript has been judged scientifically suitable for publication and will be formally accepted for publication once it meets all outstanding technical requirements.

Kind regards,

Francesca Chiesi

Academic Editor

PLOS ONE

---

## [Editor Report · Acceptance letter]

25 Nov 2020

PONE-D-20-16637R1 

Modifying and Validating the Social Responsiveness Scale Edition 2 for use with Deaf Children and Young People 

Dear Dr. Wright:

I'm pleased to inform you that your manuscript has been deemed suitable for publication in PLOS ONE. Congratulations! Your manuscript is now with our production department. 

Kind regards, 

on behalf of

Dr. Francesca Chiesi 

Academic Editor

PLOS ONE